# Using Wearable Sensors to Measure Goal Achievement in Older Veterans with Dementia

**DOI:** 10.3390/s22249923

**Published:** 2022-12-16

**Authors:** Jennifer Freytag, Ram Kinker Mishra, Richard L. Street, Angela Catic, Lilian Dindo, Lea Kiefer, Bijan Najafi, Aanand D. Naik

**Affiliations:** 1Houston VA HSR&D Center for Innovations in Quality, Effectiveness and Safety, Michael E. DeBakey VA Medical Center, Houston, TX 77030, USA; 2Interdisciplinary Consortium on Advanced Motion Performance (iCAMP), Michael E. DeBakey Department of Surgery, Baylor College of Medicine, Houston, TX 77030, USA; 3BioSensics, Boston, MA 02458, USA; 4Department of Communications, Texas A&M University, College Station, TX 77843, USA; 5Department of Medicine, Baylor College of Medicine, Houston, TX 77030, USA; 6Department of Management, Policy and Community Health, School of Public Health, University of Texas Health Science Center, Houston, TX 77030, USA; 7UTHealth Consortium on Aging, University of Texas Health Science Center, Houston, TX 77030, USA

**Keywords:** wearable, digital health, remote patient monitoring, telemedicine, dementia, Alzheimer’s disease, cognitive impairment, aging, patient goals, goal setting

## Abstract

Aligning treatment with patients’ self-determined goals and health priorities is challenging in dementia care. Wearable-based remote health monitoring may facilitate determining the active participation of individuals with dementia towards achieving the determined goals. The present study aimed to demonstrate the feasibility of using wearables to assess healthcare goals set by older adults with cognitive impairment. We present four specific cases that assess (1) the feasibility of using wearables to monitor healthcare goals, (2) differences in function after goal-setting visits, and (3) goal achievement. Older veterans (*n* = 17) with cognitive impairment completed self-report assessments of mobility, then had an audio-recorded encounter with a geriatrician and wore a pendant sensor for 48 h. Follow-up was conducted at 4–6 months. Data obtained by wearables augments self-reported data and assessed function over time. Four patient cases illustrate the utility of combining sensors, self-report, notes from electronic health records, and visit transcripts at baseline and follow-up to assess goal achievement. Using data from multiple sources, we showed that the use of wearable devices could support clinical communication, mainly when patients, clinicians, and caregivers work to align care with the patient’s priorities.

## 1. Introduction

Aligning treatment with patients’ healthcare priorities is especially challenging in dementia care [1,2,3,4]. Both patients and their caregivers face difficulties communicating to clinicians what is important to the patient and setting healthcare goals [3,4,5,6]. To set realistic and achievable outcome goals for everyone, clinicians rely on the ability of patients and caregivers to report the patient’s current health conditions, physical limitations, and barriers to goal accomplishment in validated questionnaires. While self-reported health questionnaires provide valuable information and are convenient to administer, they are subject to bias and inaccuracy [4,5].

Wearable technologies are a practical way to supplement self-reported data and remotely monitor health outcomes, such as physical activity, cognitive frailty, and sleep quality [7,8,9,10,11,12]. Wearables have been used successfully in routine care to facilitate setting goals involving mobility [5,7]. As a result, wearables can be incorporated into routine care to measure physical function. Recent research has shown that wearables can also provide insights into patients’ interactions with their surroundings and can collect data to ascertain functional and cognitive decline [2,3,11,13,14]. Over the past two decades, digital health technology to track physical activity has evolved from bulky carry-on electronics to smartphones [15,16,17] and now lightweight wearable accessories (i.e., smartwatch, body patches, or a pendant). These wearables are constantly improving in terms of battery life and firmware [18,19,20]. Furthermore, the advent of sophisticated machine learning and deep learning models has made it possible to extract different information related to physical activity [21,22]. 

These advances in the information collected by wearables can provide contextual information about physical activities and sleep quality to assess diverse healthcare outcomes. Healthcare priorities and goals are nuanced and unique to individuals, and data derived from wearables must be thoughtfully incorporated into setting outcome goals, assessing goal attainment, and revising goals based on the patient’s health status. 

The goal of this pilot study was to determine the feasibility of using wearables to assess healthcare goals set by older adults with cognitive impairment. There were three aims of the present study. First, demonstrate the feasibility of using wearables to assess healthcare goals set by older adults with cognitive impairment. Second, reveal differences in patient function after a goal-setting visit using sensor data. Last, use sensor data to assess the achievement of individual goals. The present study contributes to developing a practical approach for goal setting among patients with dementia using wearables. The present study may facilitate the integration of wearables in standard clinical care and promote effective communication among all the stakeholders involved in dementia care.

This manuscript first describes characteristics of the sample of patients who participated in this study and the feasibility of conducting the wearable sensor measurements and self-reported data collection. Then, we describe changes to wearable sensor data and self-reported measures during the two data collection periods. Finally, we provide four case examples that integrate specific patient characteristics, the focus of the patient priorities identified in each case example, and how changes in sensor measurements offer additional contexts for any treatment changes made based on the patient priorities conversations. 

## 2. Materials and Methods

The current study evaluates the potential role of wearable devices in enhancing care and expanding the potential of wearable-based outcomes. This was carried out in order to implement a goal-setting intervention among a population of older veterans with dementia. Shortly before or after their initial goal-setting visit, patients wore a pendant inertial sensor that measured physical activity and sleep quality for 48 h. They also completed self-report assessments of activities of daily living (ADLs) and instrumental activities of daily living (IADLs), as well as their function in different spaces. They then participated in an encounter with a geriatrician trained in Patient Priorities Care (PPC), a method used to identify patient health priorities and goals. Patients wore sensors 4–6 months later using the same 48 h wearable pendant and self-report assessments. 

### 2.1. Study Population

Participants were recruited from a geriatrics outpatient clinic at a Veterans Affairs Medical Center (VAMC). Veterans’ charts were reviewed to confirm eligibility criteria, including multiple comorbid conditions and whether the veteran: (a) was seen in the specialized dementia clinic, or (b) had documentation of cognitive impairment, memory loss, or a cognitive screening test indicating likely dementia. Patients meeting eligibility criteria were contacted by mail and then called either before their appointment or met at their next clinic visit to determine interest in participation. See CONSORT diagram, Figure 1. All participants provided written informed consent that was approved by the Institutional Review Board of the Michael E. DeBakey VA Medical Center (Study ID: H-43336).

### 2.2. Procedure

Patients were asked to wear a pendant sensor around their neck for 48 h or more shortly prior to or immediately after their goal-setting visit occurred. Prior studies have shown that two days of activity monitoring are sufficient to determine motor functions and frailty stages and that this period yields an optimum adherence to continuously wearing the sensor [14,23]. During the measurement period, patients and caregivers were asked to keep a paper movement diary detailing the patient’s daily activity; for example, the diary asked when the patient participated in household chores, such as cooking dinner or when the patient showered. The diary also included a record of approximate bedtime and wake-up time. Patients and caregivers were audio recorded during the visit; recordings were transcribed. The portion of the physician’s note dealing with his goals (recorded in the free text) was extracted from the patient’s electronic health record (EHR). At follow-up (an average of 5.8 months after baseline), patients were asked to wear the pendant sensor for another 48 h and keep a movement diary.

### 2.3. Patient Priorities Care

During clinic visits, physicians identified patient’s healthcare priorities with participants and their caregivers using PPC. It provides a framework for communication and decision-making during clinic visits with older adults and their caregivers [4,24]. Patients and caregivers can set specific, realistic and actionable goals, and clinicians can make informed healthcare choices aligned with patient goals (Figure 2).

PPC is effective in reducing treatment burden and is associated with a reduction in unwanted prescriptions and better alignment of self-management tasks with priorities of older adults with multimorbidity [4,5]. Incorporating PPC into clinical practice has revealed the range of personalized healthcare goals set by older adults [5]. For example, while some wish to garden daily, others simply want friends and family to come and visit. Although patients and companions can provide useful information in setting these goals, their ability to recall and report patient information is complicated by their own biases and limitations [2,3,8].

Prior to the recruitment phase of the study, the research team (consisting of clinicians, experts in communication, and implementation scientists) collaborated with participating physicians to adapt PPC for use with patients with dementia. The team used simplified language and visual aids to describe the PPC process and prompt patients and their caregivers to discuss the patient’s health priorities. The team also discussed strategies for eliciting responses from caregivers while ensuring that the encounter remained focused on the patient’s priorities. To assess the way patients responded during PPC, visits with clinicians were audio-recorded, and EHR notes from the visit were collected.

### 2.4. Wearables

A pendant waterproof inertial measurement unit (PAMSys™, BioSensics LLC, Watertown, MA, USA) was worn around the neck using a lanyard and magnetic closure (for ease of wearing and removing with minimum risk of choking) to monitor spontaneous physical activity and sleep quality (Figure 3) [7]. The PAMSys^TM^ pendant consists of a 3-axis accelerometer, battery, and built-in memory for recording long-term data. Accelerometer data are recorded at a sampling frequency of 50 Hz. Data are stored on the pendant and are downloaded and analyzed with proprietary software after use. The software uses validated algorithms to estimate cumulative postures (e.g., sitting, standing, lying, and walking), locomotion characteristics (e.g., number of steps, number and duration of unbroken walking bouts), sleep parameters (e.g., time in bed), and postural transitions (e.g., duration of sit-to-stand transitions) [25,26,27].

### 2.5. Measures

#### 2.5.1. Assessment of Dementia and Functionality

The participants’ dementia ranged in severity. To capture differences in cognitive impairment, two geriatricians who were members of the study team developed a three-category measure of cognitive impairment and a three-category measure of IADL/ADL function using scores from the Montreal Cognitive Assessment (MOCA) and the Mini-Mental State Exam (MMSE) [28,29]. The two geriatricians assigned scores using the following criteria: severe, MOCA/MMSE < 11; moderate, MOCA/MMSE 11–15; mild, MOCA/MMSE > 15. The three-category measure of function included an assessment of scores on the Katz Index of Independence in Activities of Daily Living (ADL) [30] and the Lawton–Brody Instrumental Activities of Daily Living (IADL) Scale [2]. Scores were given using the following criteria: lowest level of function, Katz = 18 or less and Lawton–Brody < 3; middle level of function, Katz = 18 or less and Lawton–Brody < 7; highest level of function, Katz = 18 or more and Lawton–Brody 7 or greater.

#### 2.5.2. Survey Measures

The World Health Organization Quality of Life measure (WHO-QOL-OLD) is a 4-question assessment of satisfaction with daily activities [31]. The Community Integration questionnaire is a 12-item measure that assesses participation in activities outside the home, visits with family and friends, and engagement in community activities [32]. The Multimorbid Treatment Burden Questionnaire (MTBQ) is a 10-item measure that assesses the difficulty of maintaining the individual’s health, including managing medications, attending medical appointments, and monitoring conditions [33]. Life-space is a measure of a patient’s ability to move about spaces, beginning at the patient’s bedroom and moving by levels to traveling out of town [34]. Life-space is scored out of 40 points, with higher points awarded for the frequency of occupying a space and occupying a space without assistance.

#### 2.5.3. Sensor-Derived Measures

Physical activity and sleep-related parameters were extracted from the pendant using two different validated algorithms. One algorithm estimated daily physical activities (e.g., cumulative postures, postural transitions, and walking characteristics), and another algorithm quantified sleep quality (e.g., sleep duration in the night, sleep onset latency). These algorithms were described in detail in the previous studies [7,35,36].

For this study, we utilized parameters to indicate physical activity, including walking duration (minutes), step count, average steps per walking bout, number of the unbroken walking bouts, which included a minimum three consecutive steps within 5 s interval [36], and percentage of walking relative to sedentary activity. We also used parameters related to sleep, including sleep duration (minutes), sleep onset latency (the length of time it takes to accomplish the transition from full wakefulness to sleep), and wakefulness after sleep onset (minutes awake after sleep begins) [36].

Sleep quality was characterized in terms of the total duration of a participant’s time in bed during the night and sleep onset latency. We described the details for extracting time in bed during the night in a previous study [36]. Briefly, a band-pass filter was first applied on the acceleration signal to reduce the unwanted noise. Then, a vector magnitude/norm of acceleration was estimated for every minute. Lastly, a model was used to estimate the sleep/wake conditions based on the moment and standard deviation calculated from every one-minute acceleration vector, posture (sleeping on slides or back), and postural transition (e.g., tossing on bed, rotating from back to slides, etc.) information. To compare physical activity at baseline and follow-up, we used a simple means comparison of 24 h of continuous wearable sensor data for each participant at time one and time two.

#### 2.5.4. Case Studies Describing the Utility of Wearable Devices

To better describe the nuanced benefits of using wearable devices, we describe case studies of PPC encounters and the related self-reported and sensor measures. Four case studies were chosen by the research team as the team examined patient profiles based on collected data, including demographics and health status, transcripts of clinic visits, sensor data, clinician notes, and questionnaire data. The first author presented complete patient cases (*n* = 14), including data from all sources to the group during team meetings, and the team discussed the selected cases in terms of the usefulness of the sensor data for measuring the effectiveness of PPC. Four cases were chosen from among the 14, each demonstrating circumstances in which sensors, visit transcripts, and self-reported measures were useful in assessing identified patient goals.

## 3. Results

Nineteen patients participated in clinic visits and wore the sensors before or shortly after their clinic visit. Fourteen (74%) patients completed the study by wearing sensors 3–6 months after their visit. The mean age of patients was 85.6 ± 6.5 years, and all were male, which is representative of the older veteran population (Table 1).

### 3.1. Feasibility of Using Sensors

Data for all but one patient (*n* = 19) could be extracted for analysis at baseline. Because the pendant device was waterproof, participants were able to wear the pendant at all times and were asked not to remove it when bathing if possible. One patient removed the pendant temporarily, which was replaced, and an additional 48 h of data were collected. Another participant removed the pendant, and it could not be found. No participants turned devices off, and none reported being bothered by the pendant or the thread and magnetic clip worn around their neck. Of the patients who wore the pendant, six (31.6%) participants were able to keep the movement diary. In some cases, caregivers were able to assist the participant in recording activity.

### 3.2. Differences in Function at Baseline and Follow-Up

We compared means for each measure generated by the pendant at both timepoints. We found no statistically significant overall differences between timepoints across the full sample (Table 2). The step count, average steps per walking bout, and percent of walking relative to sedentary behaviors trended upward, although the mean differences did not reach statistical significance. While sensor-derived measures trended upward, there was no significant change in self-reported measures. 

### 3.3. Four Case Studies Illustrating the Usefulness of Sensor Data

Patient cases were evaluated by triangulating pendant-derived data, transcripts of recorded visits, patient notes, functional assessments and survey measures. Each case was chosen to illustrate ways sensor data may supplement clinical communication with regard to identifying and aligning care with patients’ health priorities (Table 3).

#### 3.3.1. Case 1: Patient Achieved Movement Goal

**Veteran profile.** Mr. J is a 76-year-old veteran who is highly functional in ADL/IADLs but has significant memory loss. He lives with his sister, who is his caregiver. In the past, he spent time walking his dogs. However, his sister reported that he recently had less energy and interest in walking his dogs.

**PPC.** During his visit, his physician discussed the need for him to continue to engage in physical activity. Mr. J and his caregiver stated that Mr. J was no longer walking his dogs, though he had done so regularly in the past. His physician suggested that he set a goal related to walking his dogs more, and he agreed this goal was consistent with his priorities. The physician’s note in the veteran’s EHR also reflects the goal set during the visit: “He does understand the importance of remaining active and will strive to achieve this, including walking his dogs more frequently. His sister will support him in this and states that she will even accompany him to take the dogs on a walk.”

**Sensor measures.** Mr. J showed improvement in physical activity parameters. Mr. J walked more, with a higher number of walking bouts and daily step counts at follow-up than at baseline (Table 3, Figure 4). From his first measure to his second, he made the largest increases in his number of long walking bouts (+50% baseline to follow-up) and percent of walking relative to sedentary behavior (+144% baseline to follow-up). These parameters are important when considering his goal of walking his dogs regularly. An increase in long walking bouts suggests that Mr. J is taking longer walks outside of his home, as does his very large increase in walking relative to sedentary behavior. 

Mr. J’s movement diary provided more detail about his walking. He reported that he did walk his dogs in the morning while he wore the sensor. The pendant was also able to generate a timeline that captured when physical activity took place. The timeline further supports Mr. J’s report by showing long bouts of walking in the morning, which is concordant with his goal of walking his dogs in the morning.

**Survey measures.** Mr. J and his caregiver reported the lowest level possible of Treatment Burden, and his quality of life and life space measures were near the highest scores. Thus, he and his caregiver felt little treatment burden. None of the measures changed at follow-up. 

Mr. J and his clinician set a specific movement goal during his PPC visit, walking his dogs. Notably, his survey responses did not reflect any improvement or change based on this goal goals; however, the sensor reported movement consistent with his self-reported dog-walking, with long periods of walking in the morning, when he said he walked his dogs.

#### 3.3.2. Case 2: Patient Improved Overall 

**Veteran profile.** Mr. K is 89 years old and is in the lowest range of both measures of function (Table 3). He typically walks with the assistance of a rollator and lives with his wife, who is his caregiver. Mr. K has a history of coronary artery disease and is managing Type II diabetes. When Mr. K and his wife spoke to his physician, both reported that he had no falls and was sleeping well. 

**PPC.** Independence is important to Mr. K. His goal was to continue to do light housework and work in his garage. His physician’s note in Mr. K’s EHR states that it is important to Mr. K to maintain “physical strength after his two heart attacks…. He states that his goal is to maintain his current level of physical strength so that he can stay at home and prevent future hospitalizations. We discussed continuing physical therapy sessions 3 times/week and going outside at least 2 times/day, which he is currently doing.” 

**Sensor measures.** Mr. K was unable to keep a movement diary, but all parameters related to movement increased. In particular, he walked 1.5 times more at his second measurement than his first measurement. The amount of time he spent walking relative to his sedentary behavior increased nearly 1.75 times. At the first measure, Mr. K had no long walking bouts (walking > 30 steps without stopping) but had 12 during the second. Mr. K’s sleep improved in one metric—the time it took him to go to sleep after lying down decreased by over 700%. However, he had a more than 50% increase in waking up after going to sleep. 

**Survey measures.** It is notable that Mr. K’s self-reported Life-Space assessment and Community Integration scores improved significantly. These scores typically correlate with improvements in sensor-based measurements of mobility, as these measures reflect the ability to move within and outside of the home.

Mr. K improved overall in his physical activity. Although his improvement cannot be directly linked to his goal of going outside, multiple sensor parameters suggest he is likely moving about spaces more easily. For example, his long walking bouts suggests he is walking outside of his home. Additionally, improvement in his sleep is notable, although follow-up might reveal treatable reasons for the increase in waking up, such as using the restroom frequently. Mr. K’s case illustrates the benefits of combining sources of data to address the complexity of goal attainment. He was unable to report whether he was going outside more. However, pendant device measures combined with measuring outcome goals are concordant with health improvement that is expected from attainment of his goal.

#### 3.3.3. Case 3: Potential Barriers to Meeting Goals

**Veteran profile.** Mr. L is an 88-year-old man who is in the highest levels of functional categories at baseline (See Table 3). He is physically fit and walks around his neighborhood when possible. He is close with his sister, who lives in Florida. 

**PPC.** Family is important to Mr. L, and his goal was to visit his sister over the summer. Mr. L’s wife reported that his memory had recently declined, and he had become disoriented at night. However, it was important to her that Mr. L travel to Florida to visit his sister by himself so that he could spend time alone with his sister. 

During the visit, his physician recommended an alarm system for his home to make sure he did not wander at night. Mr. L’s wife discussed his disrupted sleep with his physician, stating that Mr. L sometimes went into the backyard. His physician noted that it was clear that Mr. L had disrupted sleep, but he was getting enough sleep overall. 

**Sensor measures.** Mr. L’s level of physical activity decreased significantly from baseline to follow-up. His step count decreased by a magnitude of nearly 1.5 times. His steps per walking bout and number of long walking bouts decreased by approximately 50% each. The most significant change in his activity was the amount of walking he did relative to sedentary behavior, declining by 160%. The time it took Mr. L to go to sleep improved by over 50%. However, his sleep still showed disruption. Among participants, Mr. L had one of the highest numbers of minutes awake after going to bed at 197 min (close to 3.5 h). An increase of over 10% at follow-up indicates that his wakefulness at night had grown, and he was getting less sleep than his physician noted during his visit.

**Survey measures.** Mr. L and his caregiver reported the lowest level possible of treatment burden, and his quality of life and life space measures were near the highest scores. None of the measures changed at follow-up.

Mr. L’s sleep measures support his wife’s report of sleep disturbance, and changes from baseline to follow-up indicate that the condition may be worsening. Mr. L and his wife reported no change in their self-report; however, the sensor reveals significant losses in sleep and movement. These measures highlight physical changes associated with increased confusion, which is a barrier to Mr. L’s goal attainment that may otherwise have remained unclear. Despite his wife’s desire for him to travel alone to visit his sister, the sensor data suggest that Mr. L may no longer be able to. 

#### 3.3.4. Case 4: Sensor Shows Significant Decline

**Veteran profile.** Mr. M is an 84-year-old man who lives alone. His daughter is his caregiver, and she visits him daily in his home, as do his neighbors and friends. Mr. M has limited function and needs assistance with many IADLs. He is in the lowest level of cognitive ability, with a MOCA of 10. Toward the end of his visit, Mr. M became disoriented and agitated. Much of the discussion took place between Mr. M’s daughter, who is his caregiver, and his physician. 

**PPC.** Mr. M was interested in staying at home and maintaining his independence; however, his physician and daughter were concerned about his food intake during the day. Mr. M often did not remember that he had food in his pantry that was ready to eat. His physician recommended VA homemaking services that supply light housekeeping. The homemaker could make lunch and sit with Mr. M so that Mr. M would not forget to eat. 

**Sensor measures.** Mr. M’s level of physical activity decreased significantly from baseline to follow-up. His step count decreased by a magnitude of nearly 4.7 times. He had no long walking bouts at follow-up. The most significant change in his activity is the amount of walking he did relative to his sedentary behavior, which declined by 264%. The time it took Mr. M to go to sleep improved by 35%. However, his sleep still showed disruption, with a 54% increase in waking up after going to sleep.

**Survey measures.** Mr. M’s caregiver completed the MTBQ, and her score decreased by 4 points (out of 50 points total), and her responses indicated that she considered access to healthcare services less burdensome. However, Mr. M’s quality of life and Life-Space decreased, which is concomitant with declines in his movement and sleep. 

Both survey measures and sensor data indicated a decline in Mr. M’s ability to function independently, with decreases in physical activity and interrupted sleep. As a patient becomes more dependent on his caregiver, priorities move from those of the patient to those of the caregiver. His daughter’s priority of using homemaking services to ensure that Mr. M ate lunch is a goal that serves Mr. M’s value of independence. However, his daughter may not be aware of Mr. M’s significant decline. Because of that decline, homemaking services may not provide enough support to keep him living independently.

## 4. Discussion

The results of the current pilot study show that wearable devices are feasible to use and provide useful information when implemented in combination with identifying and aligning care with the priorities of older adults with differing levels of dementia. Wearables can be feasibly used to both support self-reported information and reveal hidden information that might not be easily communicated during clinical encounters. By combining other sources of data—patient notes, recordings of visits, and survey measures—with wearable-derived data, we demonstrated that patient-caregiver-clinician communication can be supported, and even enriched, by data gathered outside of the clinic. 

Identify health priorities and aligning care with patient priorities is an important paradigm for geriatric care [25,37,38]. Providing care that enables patients to meet their goals requires more information about the way patients function outside of the clinic. However, priorities are context- and person-specific. The information needed to assess how well care aligns with priorities requires a nuanced approach to using available data (Figure 2).

In fact, the individualized nature of PPC may have contributed to the lack of measurable differences in sensor parameters across the full sample. Not only was the sample small, with varying levels of physical and cognitive function at baseline, but not all veterans set goals focused on mobility or improved function. Thus, trying to correlate PPC with functional improvement for all veterans may not be efficacious.

In Mr. J’s case, the wearable device helped measure the attainment of movement-related goals by recording physical activity. Wearables were able to characterize the type and timing of physical activity that corresponds with dog walking, and it was able to supplement future assessments of Mr. J’s goal of walking his dogs. Mr. J’s case illustrates that the wearable is especially useful in verifying goal attainment when higher-functioning patients can keep a movement diary to corroborate recorded motion. Conversely, Mr. L’s sensor data showed declines in physical activity and interrupted sleep patterns that may impede on his ability to attain his priority of traveling to see his sister. Mr. L’s decline will likely complicate his ability to successfully stay with his sister, if not prevent it entirely. Not only are there potential safety issues with his confusion and nighttime wandering, but his decreased physical function may prohibit his ability to travel alone. Moreover, as someone who sees Mr. L daily, his wife may not realize how significant declines in physical function are. She indicated that she wanted Mr. L to travel alone; using sensor data as a supplement, Mr. L’s physician could assist his wife in finding services such as an airline companion to help meet his priority. 

Mr. M’s and Mr. K’s cases illustrate ways multiple sources of data can be coordinated to supplement self-report when measuring outcome goals. Mr. K was unable to report whether he was going outside more. However, multiple measures of physical function provided by the pendant show improvement that are concordant with improved mobility. Moreover, even if Mr. K did not meet his priority of going outside more, wearable measures showed that he is capable of attaining this outcome goal. In contrast, declines in physical activity and interrupted sleep are a barrier to Mr. M’s ability to function independently. Although his daughter wanted to use homemaking services to help facilitate Mr. M’s value of independence, his daughter may not be aware that Mr. M’s decline may necessitate increased supportive services—rather than only homemaking services, Mr. M may also need services such as a home health aide, mobility assistance, or physical therapy.

Clinicians are accustomed to using data from technology to make shared decisions with patients and caregivers. Historical technological advances, going as far back as the stethoscope, have augmented the way patients and clinicians communicate [39]. Likewise, thoughtful use of wearable sensors can support communication and decision-making when clinicians tailor care to patient preferences [5,6,24,25]. They are particularly useful in the care of patients with dementia, when memory issues may impede the patient’s ability to provide important health information. Additionally, caregivers may not be aware of changes in the patient’s condition. When used to support aligning care with patient priorities, sensors can provide useful information neither patients nor their caregivers can report. 

Research involving wearable sensors for use by patients with cognitive impairment continues to focus on diagnostic and functional applications [40]. Current work in the field suggests that wearable sensors can be used in clinical practice, but it has not yet explored the use of wearables as a clinical communication tool [41]. Our pilot shows that wearable devices can support new approaches to patient-clinician-caregiver communication in the care of older adults with dementia. If multiple sources of data, including data from wearables, can be made available to clinicians to better assess care alignment with priorities, including functions not observed in the clinic setting. 

There are multiple limitations of the current study. First, most of the participants are in the later stage of dementia. Patients in the early stage of dementia may have different health priorities and may provide further insights with longer follow-up observational studies. Second, patients from the one VA geriatric clinic site were recruited. Therefore, our findings might be biased toward an organizational approach. 

In the future, a similar research study is recommended to be replicated with a similar procedure at multiple sites. Further studies using sensor data should target larger, more diverse populations in different clinical contexts. There are many more facets of wearables to explore, including optimizing sensor use for patients with varying levels of cognitive impairment, considering which wearable parameters are most relevant to specific types of goals, and designing technological tools that help clinicians efficiently use specific types of data to assess patient priorities and alignment of care with priorities.

## 5. Conclusions

As the use of wearable devices grows, the need to incorporate this data into clinical care increases. Using data from a number of sources, we have shown that thoughtful use of wearable devices can support clinical communication, particularly when patients, clinicians, and caregivers work to align care with the patient’s priorities. Therefore, the wearables may facilitate cognitive rehabilitation through longitudinal and unbiased physical activity and sleep assessment. Future studies are needed to extend the approach at multiple sites and in diverse patient population including people in the early stage of dementia.

## Figures and Tables

**Figure 1 sensors-22-09923-f001:**
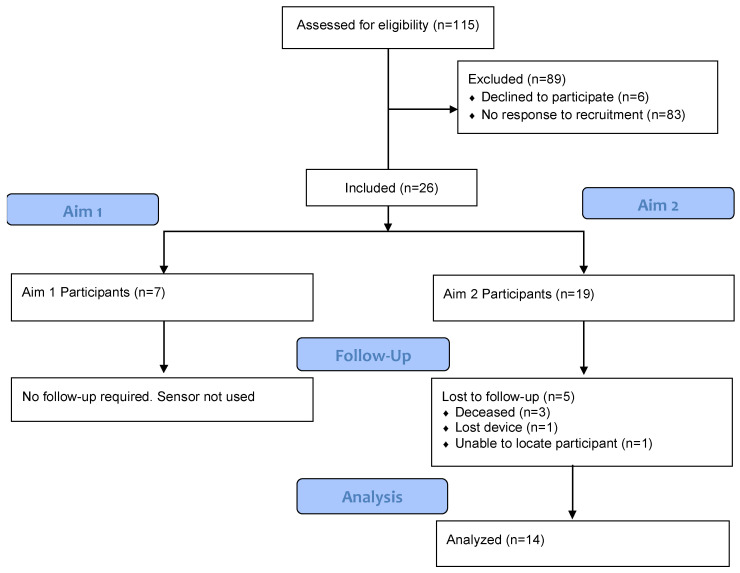
Intervention Design: Consort chart for the study.

**Figure 2 sensors-22-09923-f002:**
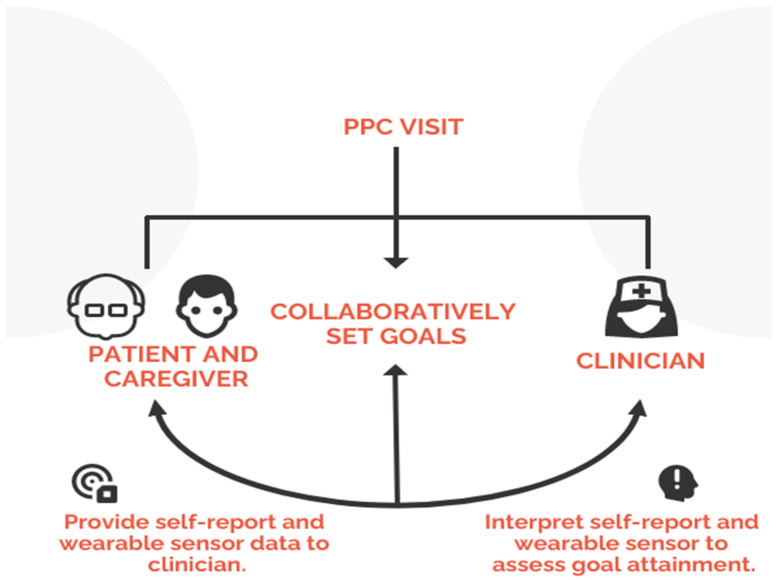
Patient Priorities Care program to determine patient-centric health priorities and setting relevant goals. Wearable sensors were used to determine patient participation in achieving the goals and facilitate the goal-setting process.

**Figure 3 sensors-22-09923-f003:**
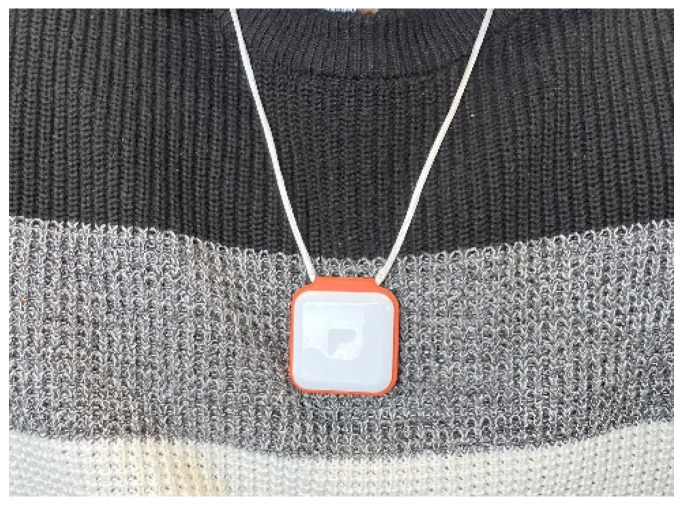
PAMSys^TM^ pendant.

**Figure 4 sensors-22-09923-f004:**
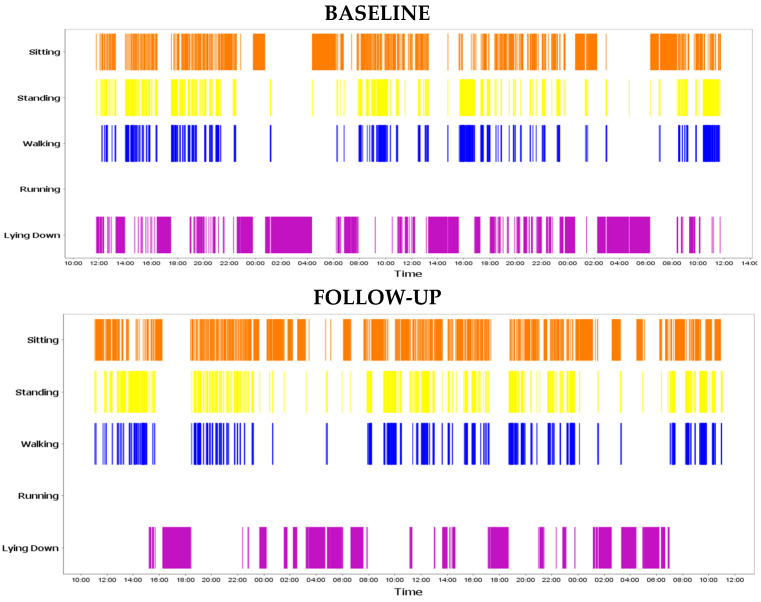
Change in activity timelines from baseline to follow-up.

**Table 1 sensors-22-09923-t001:** Demographic information.

Participant Characteristics	
Number of Participants, n	19
Gender, Male Age, Years	100% 85.6 ± 6.0
MoCA, Score	15.3 ± 6.3
**Cognitive Ability**	
Severe Impairment	7
Moderate Impairment	6
Mild Impairment	6
**Activity of Daily Life**	
Lowest Ability	8
Mid-Range	5
Highest Ability	6

**Table 2 sensors-22-09923-t002:** Mean differences in physical activity and survey measures at baseline and follow-up.

Sensor Parameter	Mean Difference	95% CI	*p*
Walking duration (min.)	8.08	−18.45, 34.60	0.536
Step count	823.45	−753.35, 2400	0.292
Average steps per walking bout ^1^	4.61	−1.42, 10.64	0.131
Number of long walking bouts ^2^	5.53	−15.29, 26.37	0.588
Percent walking relative to sedentary behavior ^3^	4.08	−11.96, 3.82	0.297
Sleep duration (min)	0.13	−0.56, 0.81	0.702
Sleep onset latency (min) ^4^	10.42	−4.35, 25.20	0.158
Wakefulness after sleep onset (min) ^5^	11.73	−59.44, 35.98	0.617
Sit-to-stand transition (seconds)	0.02	−0.37, 0.41	0.916
**Survey Measure**			
World Health Organization Quality of Life OLD (WHOQOL-OLD)	0.02	−1.88, 2.28	0.845
Multimorbid Treatment Burden Questionnaire (MTBQ)	0.67	−3.51, 3.65	0.970
Community Integration Questionnaire (CIQ)	0.73	0.98, −1.27	0.459
Life space function	−0.07	−4.61, 4.48	0.976

All measures are reported for a period of 24 h. Change is the difference between measurement at baseline and follow-up. ^1^ A walking bout is a continuous period of walking without stopping. ^2^ A long walking bout is >30 steps. ^3^ Sedentary behavior includes sitting. ^4^ Sleep onset latency is the length of time that it takes to accomplish the transition from full wakefulness to sleep. ^5^ When an individual shows evidence of wakefulness after falling asleep.

**Table 3 sensors-22-09923-t003:** Case Descriptions: All measures are reported for a period of 24 h. Change is the difference between measurement at baseline and follow-up.

	Case 1: Mr. J	Case 2: Mr. K	Case 3: Mr. L	Case 4: Mr. M
Cognitive Ability	Moderate Impairment	Moderate Impairment	Low Impairment	Severe Impairment
ADL/IADL Function	Highest Ability	Lowest Ability	Highest Ability	Mid-range Ability
Values and Goals	He enjoys his hobbies and engaging in physical activity. He was walking his dogs regularly, he but stopped. His goal is to walk his dogs every day. He is to continue treatment, and there are no issues or barriers to achieving goals.	He wants to remain as independent as possible. He would like to play with his dogs and do light housework. To maintain his strength, he will continue physical therapy weekly. He will also continue to manage his diabetes.	He enjoys spending time with his family and wants to visit his sister in Florida. He will continue treatment for memory loss. His caregiver should tell his sister that a change in environment might upset his routines. His physician also recommended alarm system because he wandered at night.	He values remaining independent and living at home. His caregiver does not live with him but visits him every morning with food. However, he sometimes does not eat lunch. His physician recommended homemaking services to prepare his lunch and ensure he eats it.
**Sensor Parameter**	**Percent change from baseline to follow-up for each sensor parameter**			
Walking duration (min.)	28	143	−19	−76
Step count	18	60	−145	−466
Average steps per walking bout ^1^	14	28	−42	−39
Number of long walking bouts ^2^	50	0 ^t^	−53	−100
Percent walking relative to sedentary behavior ^3^	144	171	−160	−264
Sleep duration (min)	1	10	0	33
Sleep onset latency (min) ^4^	8	−721	−52	−35
wakefulness after sleep onset (min) ^5^	6	56	13	54
**Survey Measure**	**Baseline Survey score, Follow-Up Survey score**			
WHO-QOL-OLD	20, 20	20, 18	16, 16	17, 16
MTBQ	2, 2	2, 4	6, 6	10, 6
CIQ	15, 15	8, 12	12, 12	13, 13
Life Space Function	30, 30	21.5, 24.5	34, 34	21, 18

^1^ A walking bout is a continuous period of walking without stopping. ^2^ A long walking bout is >30 steps. ^3^ Sedentary behavior includes sitting. ^4^ Sleep onset latency is the length of time that it takes to accomplish the transition from full wakefulness to sleep. ^5^ When an individual shows evidence of wakefulness after falling asleep. ^t^ Patient had 0 long walking bouts at baseline and 12 at follow-up.

## Data Availability

Data may be made available by contacting the first author of the study.

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
