# Peer review of "Using Wearable Sensors to Measure Goal Achievement in Older Veterans with Dementia"

_sensors, 2022, doi:10.3390/s22249923_

Round 1
Reviewer 1 Report
Very interresting pilot study.
However, the paper does not seem to me suitable for the journal. Indeed, to the reader of it, there is a lack of technique. The hardware and software aspects are completely absent. Emphasis is placed on empirical results that are perhaps more suitable for a medical journal.
Some comments are put in the attached pdf.

Reviewer 2 Report
The publication presents a well-written study on wearable sensors for measuring goal achievement with adults experiencing cognitive impairment. The work is well-documented and I don't see anything specific in methodology or results to change.
As an outlook, one question is what to do with the results. Since a simple motion sensor was used, but required data to be downloaded before analysis, what is the feasibility of translating the findings to existing wearable sensors and smartwatches with cloud connectivity? Can existing motion values from devices such as the Fitbit and Apple Watch family be used to scale the solution in future work?
Reviewer 3 Report
Summary: The purpose of this study was to show that employing wearables to measure healthcare goals set by older persons with cognitive impairment is feasible. The authors describe four cases that evaluate (1) the practicality of utilizing wearables to track healthcare objectives, (2) functional differences after goal-setting visits, and (3) goal attainment. The authors demonstrated, using data from many sources, that the adoption of wearable devices could assist clinical communication, particularly when patients, physicians, and carers collaborate to align care with the patient's priorities.
Strong Points: 1. The theme considered in this work is quite interesting and covers a vital area for modern societies 2. The article is well-structured and easy to read 3. The feasibility of the suggested approach is demonstrated by experimental results. Comments and Suggestions: 1. This work's primary limitation is the small number of patients—only four patients—on whom experiments were conducted. The authors must provide strong justification for this fact. 2. The authors should include a section on related work where they discuss how their work compares to earlier, similar contributions in the literature. 3. A table summarizing the related work section will be of great value too. 4. The general layout of the article is not very neat. The authors need to improve the quality of the layout. For example, a table and its title must not appear on different pages. 5. Some sections, such as Section 3.2, are too brief. 6. The authors may enrich the article with some photos of the wearables they used in their study. 7. In the introduction, the authors may make a list outlining their key contributions. 8. In addition, the authors may include a small paragraph that describes the structure of the paper. 9. The text in Figure 1 appears to have a typo (word underlined with a double blue line). 10. The authors need to discuss the possibility of using the sensors of smartphones for collecting data and supervising patients. The following references explain how this can be done. The authors are invited to include them in their study. - https://ieeexplore.ieee.org/abstract/document/9471869 - https://ieeexplore.ieee.org/document/9327468 - https://www.mdpi.com/1660-4601/18/23/12652 11. To broaden the scope of the paper, the authors may add the following references in the first and second paragraphs of the introduction: - https://www.sciencedirect.com/science/article/abs/pii/S2210670720307903 - https://link.springer.com/article/10.1007/s11063-020-10414-5 - https://www.mdpi.com/journal/sensors/special_issues/WMS_AI - https://www.mdpi.com/journal/sensors/special_issues/wearables_AI 12. Wearable sensors are being used more and more in fields like healthcare, agriculture, and the smart grid. With recent technological advancements, the development of wearable technology is progressing. However, achieving power effectiveness, hardware complexity reduction, and configurability is challenging. It is recommended that authors refer to the following articles in this context: - https://doi.org/10.3390/s21041511 - https://doi.org/10.1016/j.bbe.2022.05.006 - https://www.mdpi.com/1424-8220/21/16/5589/htm - https://www.mdpi.com/2075-4418/12/10/2563/htm 13. It is necessary to indicate the limitations of the proposed approach and the cost of the material used in this work. 14. The conclusion is not long enough. It should be expanded, and possible future work directions should be added.
Round 2
Reviewer 1 Report
Ok
Reviewer 3 Report
The authors took into account all my comments and suggestions. Good luck.